# Fluviograph Design Based on an Ultra-Small Pressure Sensor

**DOI:** 10.3390/s19214615

**Published:** 2019-10-23

**Authors:** Shanshan Li, Qingming Duan, Xunyu Chu, Chao Yang

**Affiliations:** College of Instrumentation & Electrical Engineering, Jilin University, Changchun 130026, China; liss6515@mails.jlu.edu.cn (S.L.); chuxy6515@mails.jlu.edu.cn (X.C.); yangchao18@mails.jlu.edu.cn (C.Y.)

**Keywords:** groundwater dynamic monitoring, water pressure, atmospheric pressure, water level, water temperature

## Abstract

Groundwater dynamic monitoring of assessment points and evaluation areas has a significant predictive effect for controlling the occurrence of disasters. Obtaining water level and water temperature change data can provide important theoretical significance and reference values. However, in some remote areas of China, the measurement data concerning water level change are mostly obtained by manual measurement. This measurement method not only wastes manpower, but also cannot ensure the accuracy and real-time nature of the data. Therefore, this paper carried out research and design on a fluviograph, based on the relationship between hydraulic pressure and water depth. In the paper, the fluviograph used ultra-small pressure sensors to complete the data acquisition of the water level, a STM32L011 single-chip microcomputer (STMicroelectronics, Geneva, Switzerland) to process the data, and LabVIEW software to display the final data. Additionally, the water level data record and water temperature information record can be fed back to the user and the manager. After laboratory testing, the water level variation error range of this fluviograph was 1–2 cm, and the water temperature error range was less than 1 °C, which indicates the accuracy of the metrical data. The results show that the fluviograph realizes the function of automatically recording the water level and water temperature of the monitoring point, and it improves the social production efficiency greatly.

## 1. Introduction

In the arid regions of northwest China, the lack of precipitation, high-intensity sunshine, and large-scale evaporation have caused severe conditions, including insufficient water resources, unstable ecosystems, and unbalanced development of water and soil resources [1]. Groundwater, as an important component of freshwater resources [2], is an ideal source of water supply [3,4] due to its excellent water quality, wide distribution, and reliable stability. In some semi-arid and arid regions of north and northwest China, groundwater is the primary, and sometimes even the only, source of domestic water and industrial water [5,6,7]. However, with the increasing number of wells and increasing water consumption [8,9], some over-exploited areas have experienced a continuous decline in regional groundwater levels [10,11,12], resulting in diverse geological catastrophes [13]. In order to control the occurrence and development of these disasters, people need to grasp the groundwater dynamics on a large scale and make specific assessments of their social impacts. Above all other measurement parameters, groundwater level monitoring is the most basic and important part of groundwater monitoring [14].

Existing fluviographs can be broadly classified as float type, ultrasonic type or radar type [15,16,17]. However, float gauges have large cumulative measurement errors and require frequent recalibration. The accuracy of ultrasonic instruments is affected by environmental factors such as temperature and humidity. Radar instruments are expensive, and their measurement process is affected by raindrops and snowflakes. Since video surveillance has become the standard configuration for metering stations, image-based water level measurement technology has received more and more attention in recent years [18]. Instead of the human eye, an image of an individual’s meter can be captured with a camera and automatically processed to detect the reading of the water line [18]. However, due to the low imaging resolution, tilting of the viewing angle, and complex lighting in the field, it is difficult to determine the readings on the worker’s meter.

Fluviographs based on fiber Bragg grating (FBG) pressure sensors or piezoresistive pressure sensors are used to measure the water level. FBG is becoming more and more popular in the sensor field because of its small size, corrosion resistance, anti-interference and easy reuse [19]. The Fiber Bragg Grating pressure sensor has attracted more and more attention in water level detection and measurement. However, the pressure coefficient of bare grating is only 3 pm/MPa [20], which is relatively low and far from the actual measurement index. In addition, FBG can induce pressure and temperature simultaneously, which can easily cause the problem of interleaving sensitivity in the process of acquisition and measurement [21,22]. A typical FBG pressure sensor is based on a double grating structure [23,24,25]. One grating simultaneously senses pressure and temperature changes, and the other grating is used to provide a temperature compensation coefficient sensitive only to temperature. However, in the actual water level measurement process, the pressure and temperature of the measuring environment change continuously and irregularly. For the typical double grating structure, a grating which is sensitive to temperature and pressure responds to temperature change much faster than the grating sensitive only to temperature change. Therefore, it takes a long time for the sensor to detect that the temperature of the two gratings is the same and to obtain the temperature compensation coefficient. It is difficult for the typical double grating structure to measure accurately in a rapidly changing pressure environment. The piezoresistive pressure sensor is composed of an elastic pressure sensitive component and a conversion component. The pressure-sensitive component usually adopts an elastic flat diaphragm and four diffusion resistors as conversion components. The four diffusion resistors constitute a Wheatstone bridge and place stress on the diaphragm. The change is converted to a voltage signal output [26]. Since the pressure change signal and the temperature signal perceived by the piezoresistive pressure sensor interfere with each other, they are nonlinear in nature [27,28]. This requires pressure to compensate for temperature and vice versa [29]. In the temperature compensation algorithm, the curve fitting method or linear interpolation method, and so on, can be used, but these methods easily to lead to “under-fitting” or “over-fitting”, low adaptability and low precision of test results. A neural network algorithm can also be used. However, this kind of trained algorithm requires a large amount of processing power to calculate, which reduces its practicality. In addition, if the Wheatstone bridge is simply applied to the water level detection, it is necessary to consider not only the accuracy but also the shape processing and sealing problems.

The aim of this paper is to design a low-power, high-precision and stable fluviograph. The fluviograph can automatically record the water level of the water level detection point and simultaneously detect the water temperature at the point, and can feed back the water level data and water temperature information to the user and the management personnel. By observing changes in the water level, it is possible to control unnecessary disasters. It is no longer necessary to artificially measure the water level, reducing labor and improving efficiency.

In order to control a natural disaster caused by a change in water level, the purpose of designing the water level monitoring system is proposed. Firstly, using the principle of pressure measurement of the water level, the overall design of the fluviograph is proposed. The overall software design of the fluviograph and the design of each subroutine software are given, including the system’s main program design, real time clock (RTC) design, sensor data acquisition design, electrically erasable programmable read only memory (EEPROM) design and upper computer LabVIEW program design. Secondly, a laboratory simulation test is performed on the system to verify its functionality, with error analysis and correction of the results to reduce the error. Other methods are compared with the method of this design to illustrate the advantages of choosing the pressure sensor MS5837. Finally, a summary analysis of the entire design is presented, mentioning the advantages of the overall design and pointing out possible future work. This study used ultra-small pressure sensors for pressure testing and made the pressure changes and water level changes converge. In this case, the fluviograph can meet the design requirements of low cost, low power consumption, high accuracy and easy operation.

## 2. System Design

This section describes the design of the various modules of the fluviograph. Firstly, the principle of measuring the water level using pressure is explained. Secondly, the design of the lower computer is described. The overall design of the fluviograph, the design of the front-end acquisition module and the data processing are introduced. The software design of the entire system and sub-modules is also described in detail. Finally, the design of the host computer is detailed. Mainly, the software design of the upper computer and the user interface mean that the administrator can intuitively monitor the data.

### 2.1. Principle of Water Level Measurement

This design involves the subtraction of atmospheric pressure and water pressure, so two sets of pressure detection devices were designed, namely, an atmospheric pressure detection device and an underwater pressure detection device. In real life, the atmospheric pressure changes every time, since it decreases as the height of the position increases. In addition, changes in the temperature and humidity of the air also influence atmospheric pressure [30,31,32], which decreases when temperature increases.

Therefore, the basic principle of this design is to measure the water level based on the relationship between water pressure and water depth. When the pressure sensor is connected to the cable rope and they are placed together to a certain depth in the well, the relationship between the water pressure of the sensor and the water depth is:(1)$$P=\rho \times g\times H+P_{0}$$
where *P* is the water pressure measured by the sensor, $P_{0}$ is the atmospheric pressure on the liquid surface, ρ is the density of water, g is the acceleration of gravity, and *H* is the depth of water.

When the pressure sensor with the waterproof cable rope connected is put into a certain depth of water in the well, according to Equation (1), the water level *H* = (*P* − *P*_0_)/(ρ × g) can be calculated. Among them, (ρ × g) is known.

Taking into account the relationship between temperature and water density, it is necessary to use the temperature compensation method in order to obtain a more accurate and stable pressure value. According to scientific research and theoretical experience, the relationship between water density and temperature [33], which was obtained using the polynomial regression method, can be given as:(2)$$\rho_{w}=-0.000003T_{w}^{2}-0.000108T_{w}+1.000937$$
where $\rho_{w}$ is the water density, and *T_w_* is the water temperature.

Based on the above theoretical analysis, as long as *P* and $P_{0}$ are obtained, the water level data can be calculated. The first step is measuring the water pressure *P*, the atmospheric pressure $P_{0}$, and the temperature of the water at the monitoring point at different monitoring sites. The second step is processing the data according to Equation (2). This step is in order to perform temperature compensation according to the influence of pressure on the change in temperature. The third step is using the single-chip computer to process the pressure and temperature data to obtain the final water level value. At the same time, the single-chip computer will record the water level and water temperature automatically.

### 2.2. Overall System Design

The overall system design diagram is shown in Figure 1. This system was composed of several sub-modules, namely, a front-end sensor data acquisition module, the main controller STM32L011, an electrically erasable programmable read only memory (EEPROM) memory module, a serial communication module, and a LabVIEW data processing and display module.

After powering on the system, the real time clock (RTC) was used in STM32L011 [34] to type in the corresponding command, and the data acquisition time of the sensor was set. After the sensor had responded, the data acquisition time interval was set. After this setting up, the front-end MS5837 pressure sensor data acquisition module began to collect pressure data and temperature data. The main controller STM32L011 stored the RTC time, pressure data and temperature data collected by the sensor and stored them to EEPROM. Then, the data were transferred to LabVIEW through the serial communication module, and further data processing was performed to obtain water level data. Water level changes were recorded and displayed, so that managers could know the information about water level and water temperature.

### 2.3. Front-end Sensor Data Acquisition Module Design

#### 2.3.1. Pressure Sensor Selection

MS5837-30BA is manufactured by TE Connectivity (TE) (Schaffhausen, Schweiz) [35]. It is an ultra-small gel-filled pressure sensor with 0.2 mbar high resolution, 2 mm water depth resolution and fast analog-to-digital signal conversion down to 0.5 ms. The sensor module includes a high-linearity pressure sensor and ultra-low power 24-bit sigma-delta analog-to-digital converter (ADC) with internal factory-calibrated coefficients. The six coefficients that needed to compensate the technological changes and temperature changes are calculated and stored in the 112-bit programmable read only memory (PROM) of each module. Therefore, the pressure sensor can measure pressure and temperature accurately and stably.

#### 2.3.2. Data Acquisition Program Design

MS5837 follows the inter-integrated circuit (I2C) communication protocol. The external microcontroller clocks in the data through the input serial dlock (SCL) and serial data (SDA). The sensor responds on the same pin SDA, which is bidirectional for the I2C bus interface. So, this interface type uses only two signal lines.

The temperature and pressure calculation process is performed in two steps according to the datasheet [35].

The first step is to read the calibration data C1–C6 from PROM, where, C1 is the pressure sensitivity coefficient (SENST1), C2 is the pressure offset coefficient (OFFT1), C3 is the temperature coefficient of pressure sensitivity coefficient (TCS), C4 is the temperature coefficient of pressure offset coefficient (TCO), C5 is the reference temperature coefficient (TREF), and C6 is the temperature coefficient of the temperature (TEMPSENS). The digital pressure value D1 and the digital temperature value D2 are read according to the sequential command, then the temperature is calculated. The calculation formula can be given as:(3)$$dT=D2-C5\times 2^{8}$$
(4)$$TEMP=2000+dT\times C6/2^{23}$$
where *dT* is the difference between the actual and reference temperatures, and *TEMP* is the actual temperature.

The calculation formulas of the temperature compensated pressure, *OFF*, *SENS*, and *P*, can be given as:(5)$$OFF=C2\times 2^{16}+\left(C4\times dT\right)/2^{7}$$
(6)$$SENS=C1\times 2^{15}+\left(C3\times dT\right)/2^{8}$$
(7)$$P=(D1\times \frac{SENS}{2^{21}}-OFF)/2^{13}$$
where *OFF* is the offset at actual temperature, *SENS* is the sensitivity at actual temperature, *P* is the temperature-compensated pressure.

The second step is to perform temperature compensation [36] and to calculate the compensated temperature and pressure values. The main task is to judge the temperature value. First, determine whether the temperature is less than 20 °C, and if the temperature is lower than 20 °C, then the temperature compensation formulas are as follows:(8)$$Ti=3\times dT^{2}/2^{33}$$
(9)$$OFFi=3\times {\left(TEMP-2000\right)}^{2}/2$$
(10)$$SENSi=5\times {\left(TEMP-2000\right)}^{2}/2^{3}$$

If the temperature is higher than or equal to 20 °C, then the temperature compensation formulas are as follows:(11)$$Ti=2\times dT^{2}/2^{37}$$
(12)$$OFFi=1\times {\left(TEMP-2000\right)}^{2}/2^{4}$$
(13)SENSi=0

Finally, the higher-precision temperature and pressure values can be obtained by Equations (14) and (15).
(14)$$TEMP2=\left(TEMP-T_{i}\right)/100$$
(15)$$P2=(\frac{\frac{D1\times SENS2}{2^{21}}-OFF2}{2^{13}})/10$$
where $T_{i}$ is the compensated temperature parameter, *TEMP*2 is the final high-precision temperature value, and *P*2 is the final high-precision pressure value.

### 2.4. System Software Design

#### 2.4.1. Software Design Framework

After the fluviograph is powered on, the software program will initialize the single-chip microcomputer STM32L011 and set the data acquisition time and data storage interval through RTC in STM32L011. When the data acquisition time is reached, the pressure sensor will collect the pressure data and temperature data, and then transmit the collected data to the single-chip microcomputer STM32L011 through the I2C communication protocol. After the data processing in STM32L011, the precise pressure value and temperature value can be obtained.

The fluviograph does not collect and store data at all times. It starts collecting and storing when the set RTC time interval is up. When the data have been collected and stored, the fluviograph enters stop mode and waits for the next time interval to arrive. When the fluviograph works in stop mode, the current consumption is 665 nA, which enables low power consumption.

When the reset button is pressed, the system resets. Then, read memory command is sent, and the number of record points to be read is set. The STM32L011 transfers the pressure data and temperature data to the LabVIEW upper computer through the communication module. Through data processing in LabVIEW, the water level data can be achieved. The software flow chart is shown in Figure 2.

#### 2.4.2. Real Time Clock Subroutine Design

The internal RTC of the STM32L011 is a separate binary-coded decimal (BCD) timer/counter. The RTC clock source for this design is an external crystal oscillator of 32.768 KHz. Since the fluviograph involves synchronous data acquisition of over water and underwater devices, the data acquisition time should be synchronous so that the two devices can start collecting at the same time, which facilitates data processing and ensures the correctness of the results. In addition, the atmospheric pressure changes all the time; it changes with time, place, temperature and other factors. Using the RTC alarm mode to set the data acquisition interval time can allow the realization of multi-time measurement, which reduces the error.

#### 2.4.3. EEPROM Memory Subroutine Design

AT24C512 is a serial electrically erasable and programmable read-only memory [37]. It follows the I2C protocol. The correct writing of the device address, write operation and read operation sequence lays a certain foundation for the successful realization of the corresponding functions of memory.

In the EEPROM memory software programming, the EEPROM should be initialized first, that is, set the SCL and SDA pins’ initial values; secondly, determine whether the 24C512 chip is detected. When the detection is successful and the sensor has collected the data within the time interval, write the pressure data and temperature data into the EEPROM memory; finally, when the main controller issues a read data command, the data are read from the EEPROM according to record points set by the user.

### 2.5. Upper Computer LabVIEW Program Design

The upper computer LabVIEW provides the user with a visual and graphical operation interface [38]. The data processing interface is shown in Figure 3. It includes a serial port configuration unit, a sensor calibration time setting unit, a sensor data acquisition time setting unit, a temperature and water density correction unit, a water level data storage unit, and a data display unit.

The flow chart of the LabVIEW upper computer design is shown in Figure 4.

The function of the serial port configuration unit is to select the serial port number and the communication baud rate. The function of the sensor calibration time unit is to obtain the current Beijing time and the correction time. The sensor data acquisition time setting unit is configured to set a sampling time interval and start data acquisition time. The water level information storage unit mainly stores the atmospheric pressure and the water temperature in the form of a .txt file in the corresponding folder. The data processing function of the sensor in the air is the same as that of the sensor underwater.

The data display unit performs the subtraction of the pressure data of water pressure and atmospheric pressure, and displays the time information, temperature and water level information in a spreadsheet format. Then, these values are stored in .txt format to the folder under the specified path. This means that the water level calculation is completed and the water level changes are recorded.

## 3. Results and Discussion

In this part, the fluviograph was simulated and the temperature and water level changes at different times were obtained. The atmospheric pressure-detecting device and the hydraulic pressure-detecting device were subjected to error calibration. Comparing the pressure sensor MS5837 with a FBG pressure sensor and a piezoresistive pressure sensor highlights the advantages of choosing the MS5837 as an acquisition module. The fluviograph’s power consumption is very small, meeting the requirements for low power consumption. The experimental results were analyzed and summarized, and the fluviograph was shown to realize the function of automatic detection.

### 3.1. Experimental Results

In this paper, a standard scale transparent water pipe was used as a groundwater level simulation device and a standard thermometer was used for the temperature test to verify whether the function of this design could be realized. The transparent water pipe had a height of 1 m and a scale of 1 cm. By changing the position of the sensor probe in the water pipe, the real-time water level changes could be measured.

The data acquisition interval was set to 5 s. The instrument started to collect at 9:09:00 on 26 May 2019. The water level was 5 cm, the atmospheric temperature was 25 °C, and the water temperature was 26.6 °C. The test data are shown in Table 1 and Table 2.

### 3.2. Error Calibration Method

This design requires the pressure value $P_{0}$, measured by the atmospheric pressure-detecting device and the pressure value *P*, measured by the hydraulic pressure-detecting device, to be subtracted. This means that when the two are measured in the same environment (in air or in water), the values are exactly the same. We measured both sets of inspection devices in the atmosphere and in the wells at the same time. After many tests, we found that the results of multiple measurements in the same environment at different times were different by 20 mbar, that is, the initial error was Δ = 20 mbar. The pressure measured by the atmospheric pressure-detecting device is always 20 mbar less than the pressure measured by the hydraulic pressure-detecting device. This requires the pressure value of one set of equipment to be the standard value, and the pressure value measured by the other set of equipment is then calibrated according to the standard value. When the result of the hydraulic pressure-detecting device is used as a standard value, the atmospheric pressure-detecting device is calibrated, and the corrected result is $P_{0}$’ = $P_{0}$ + Δ.

### 3.3. Temperature Compensation

The pressure and temperature of the measurement environment change continuously and irregularly. A typical fiber grating pressure sensor tries to detect whether the temperatures of the two gratings are consistent and whether the temperature compensation coefficient has been obtained, which takes a long time. Piezoresistive pressure sensors need to use the curve fitting method, linear interpolation method and other methods for temperature compensation, but these methods easily to lead to “under-fitting” or “over-fitting”, low adaptability, and low accuracy of test results. For MS5837, the sensor integrates multiple factory calibration compensation coefficients, and the six coefficients required to compensate for process variations and temperature changes are calculated and stored in the 112-bit PROM of each module. Therefore, using the MS5837 pressure sensor as an acquisition module can reduce many complicated operations and improve the designer’s work efficiency. In addition, its analog signal conversion speed is as low as 0.5 ms, which can quickly sense pressure and temperature changes, thus accurately and stably measuring pressure and temperature.

### 3.4. Low Power Consumption

The fluviograph uses a 3.7 V, 1800 mAh rechargeable lithium battery as the power supply voltage, and the system operates at 32.768 KHz. When the system is in run mode, the total current consumption of STM32L011 and all peripherals is 29.64 mA within 1 ms. As shown in Figure 5a, the battery life can be up to six years. When the system is neither collected nor stored, the system is in stop mode, the total current consumption is 665 nA, and the current consumption is shown in Figure 5b. The STM32L011 and MS5837 used in this design are low-power products, so the entire system consumes very little current, whether in run mode or in stop mode, ensuring the longevity of the system’s working time, extending the life of the system, and meeting the low power requirements. This allows the fluviograph to monitor water level changes for extended periods of time.

### 3.5. Discussion

According to the above test results, the water level recorder can effectively display the water level and water temperature change information from the well. The water level error is about 1 cm, and the temperature error is less than 1 °C; the error range is small, which realizes the automatic detection function of the fluviograph and ensures the accuracy of water level data and water temperature information.

The fluviograph can meet the needs of measuring the water level change of the well, but if the fluviograph is put into mass production and is provided to residents in order for them to use the well, teaching those non-technically adept individuals how to install the equipment and how to operate the equipment will become a problem. Therefore, further improvements are needed in terms of equipment stability and simplification of operation methods. This fluviograph system has been tested several times, and two problems still need to be solved.

First, when the system reads the number of recording points in the memory, the reading speed is slow. Meanwhile, through increasing the baud rate to speed up the storage of data, data transmission accuracy will be reduced.

Second, for convenience, a wireless communication function can be added. However, this fluviograph design requires long-term detection to observe the water level change, so the battery power consumption issue has to be considered. Considering how high the power consumption of the wireless communication function can be, this design does not involve any wireless communication function.

## 4. Conclusions

This is a laboratory proof-of-concept of a sensor; for field applications it will likely require some calibration and additional testing. This design uses a STM32L011 microcontroller as the main control chip, combined with a MS5837-30BA ultra-small gel-filled pressure sensor to collect pressure and temperature data. Data are transferred to LabVIEW through the serial communication module. After the data processing of the upper computer, the function of detecting the water level change and displaying the temperature change in the water is realized. The design is characterized by miniaturization, economy, and simple operation. The communication module is stable and reliable, and the display interface is friendly and easy to understand, so that the monitoring personnel can monitor the operation of the system in real time. This design could offer good application value in many different fields, such as groundwater exploitation monitoring points, rural water wells and daily water storage system development, which will bring convenience to people’s lives and industry.

We will optimize the fluviograph, perform multiple tests and check the error on different occasions, to confirm that the fluviograph can monitor the water level stably and accurately in many fields, and can be utilized in a wide range of production applications. We will build a network detection platform or mobile App (application), and use more advanced communication technology to transmit the data of the lower computer to the network detection platform or mobile App.

## Figures and Tables

**Figure 1 sensors-19-04615-f001:**
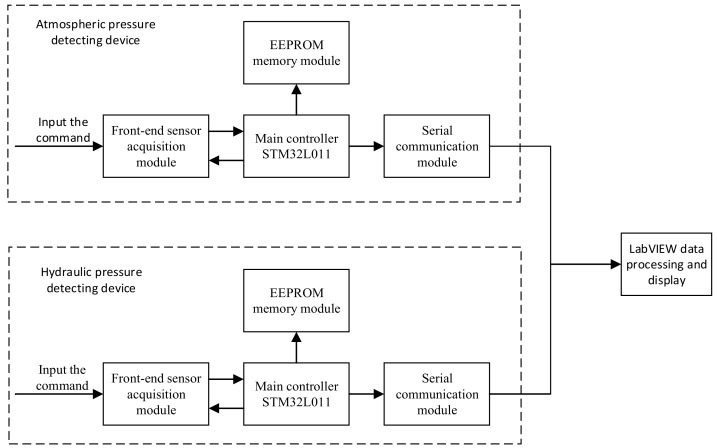
Diagram of the overall program design. The atmospheric pressure-detecting device was used to detect the pressure and temperature in the atmosphere, and the hydraulic pressure- detecting device was used to detect the pressure and temperature at a certain depth in the well. The two detecting devices and LabVIEW constituted the fluviograph.

**Figure 2 sensors-19-04615-f002:**
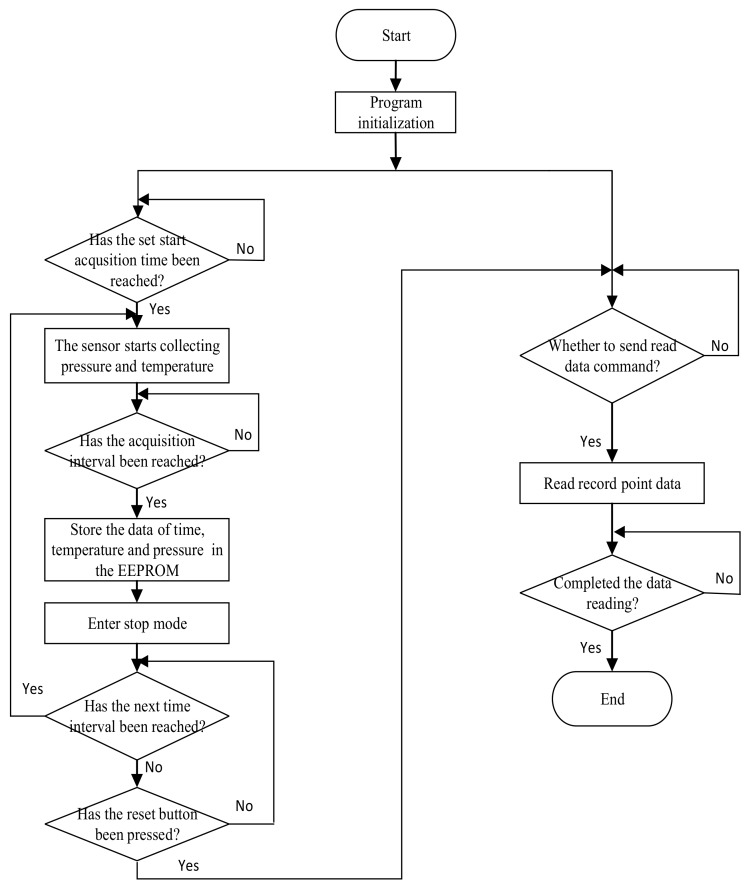
System software flow chart. Real time clock (RTC), MS5837 and electrically erasable programmable read only memory (EEPROM) software design steps and data processing.

**Figure 3 sensors-19-04615-f003:**
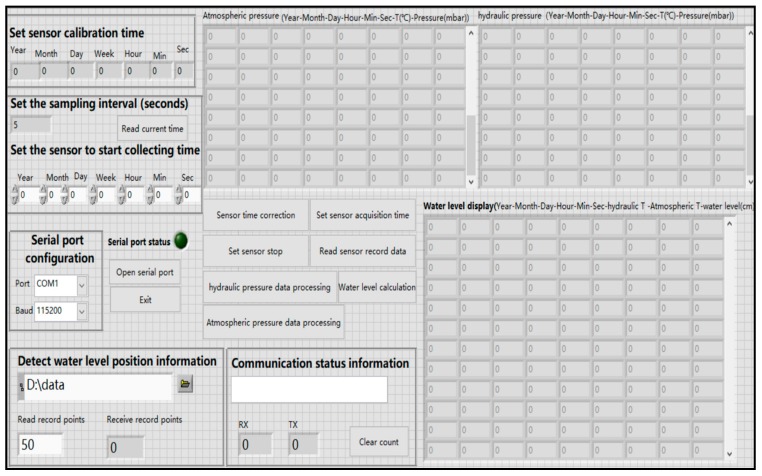
Parameter setting for water level detection and software display interface for data record recovery.

**Figure 4 sensors-19-04615-f004:**
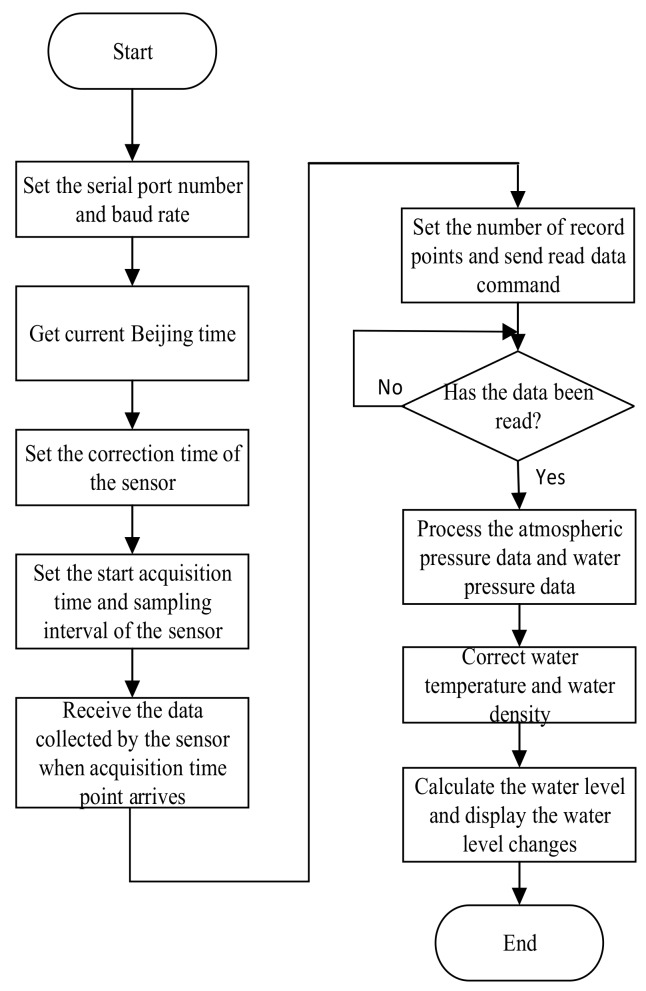
LabVIEW host computer design flow chart. The steps of parameter settings and the formation process of the display interface.

**Figure 5 sensors-19-04615-f005:**
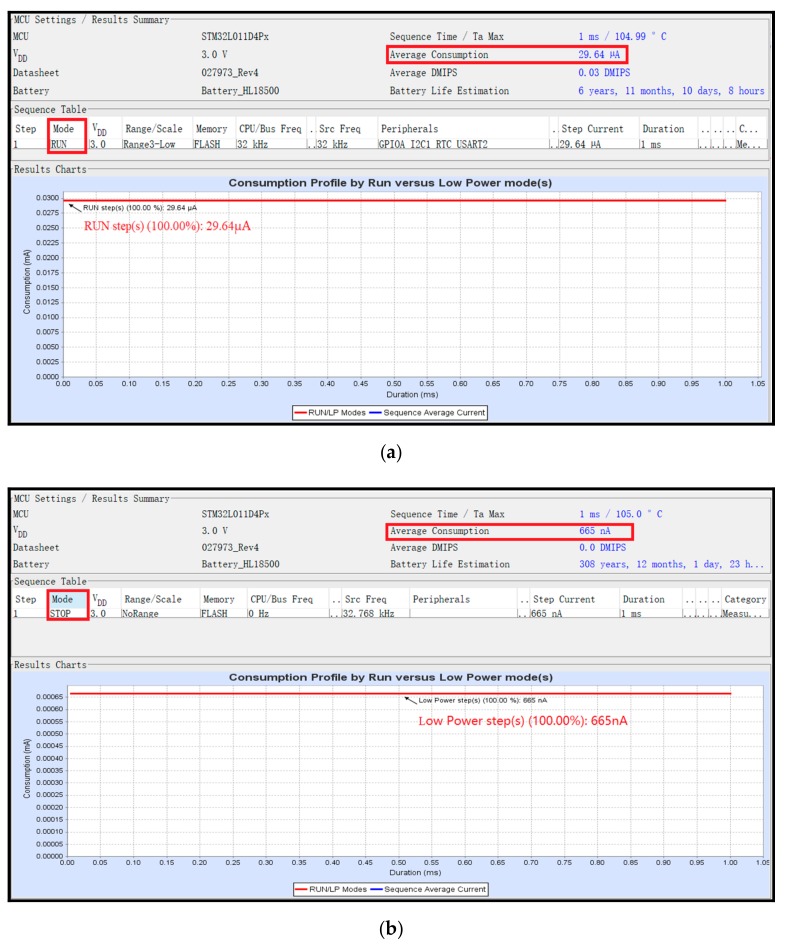
The power consumption current of the water level gauge in run mode and stop mode: (**a**) the power consumption current of the system in run mode; (**b**) the power consumption current in stop mode.

**Table 1 sensors-19-04615-t001:** Water temperature and air temperature data measured by the fluviograph.

Equipment Detection Time	Standard Water Temperature (°C)	Tested water Temperature (°C)	Error (°C)	Standard Air Temperature (°C)	Tested Air Temperature (°C)	Error (°C)
9:09:00	26.6	26	0.6	25.1	25	0.1
9:15:00	26.4	26	0.4	25.3	25	0.3
9:20:00	26.1	26	0.1	24.2	24	0.2
9:24:00	25.8	25	0.8	24.3	24	0.3
9:34:00	25.6	25	0.6	24.5	24	0.5
9:44:00	25.5	25	0.5	24.7	24	0.7
9:54:00	25.5	25	0.5	25.0	25	0
10:04:00	25.4	25	0.2	25.2	25	0.2
10:14:00	25.2	25	0.4	25.2	25	0.2

**Table 2 sensors-19-04615-t002:** Results of water level change detected by the fluviograph.

Equipment Detection Time	Actual Water Level (cm)	Tested Water Level (cm)	Error (cm)
9:09:00	5	5	0
9:15:00	9	9	0
9:20:00	15	15	0
9:24:00	23	22	1
9:34:00	42	41	1
9:44:00	50	50	0
9:54:00	65	65	0
10:04:00	80	81	1
10:14:00	92	92	0

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
