# Peer review of "Fluviograph Design Based on an Ultra-Small Pressure Sensor"

_sensors, 2019, doi:10.3390/s19214615_

Round 1
Reviewer 1 Report
“Mastering the data of water level and water temperature change can provide significant theoretical significance and reference value for monitoring groundwater dynamic monitoring in monitoring sites and evaluation areas.” This sentence is unclear; please reword and indicate what is meant by “mastering.”
“When the pressure sensor is connected to the cable and placed together to a certain depth in the well.” Would it not suffice to indicate that the sensor is placed to a given depth in the well? It can be indicated here that the cable provides communications and power to the sensor.
“What requires special explanation is that the density of water changes with temperature. When the temperature lies in the range of 0°C to 4°C, water follows the law of heat shrinkage and cold expansion, also known as "abnormal expansion". Which means, as the temperature rises, the volume of water becomes smaller and the density of water becomes larger at the same time. When the temperature is higher than 4°C, the density of water decreases as the temperature rises. In general, as the temperature rises from 0°C the density of water becomes larger at first and then becomes smaller. Nevertheless, the atmospheric pressure always becomes lower as the temperature rises.”
This paragraph is unnecessary to the exposition of the text. Please remove.
For Equation (2) indicate where the data was obtained. Is this equation from a reference paper?
Section 2.2. Remove the words “and so on” since this is not descriptive.
Section 2.2 is written in the wrong tense. Please re-write using simple past tense.
For MS5837-30BA, please indicate the manufacturer and a datasheet can be cited here.
Section 2.3.2 can contain a citation to the datasheet or application note that deals with this sensor.
Section 3.1. Please re-write all of this section in the past tense.
For Table 2, what are the significant figures to which these measurements are reported? Can the calculus-based method of error calculations be used with Equation (1) to quantify error in the measurements?
In the Conclusions section, please state that this is a laboratory proof-of-concept of the sensor and that field applications will likely require some calibration and additional testing. The emphasis on laboratory testing should also be indicated in the abstract.
Reviewer 2 Report
The paper presents the use of pressure sensors for monitoring water depth in wells. The authors use commercial sensors and most of their efforts are done in having a good system description section. However, the paper has serious shortcomings in the other sections. In its current status this paper cannot be published in a journal like Sensors. Following, I include some recommendations to enhance the quality of the paper.
The introduction must be seriously extended. First, the authors must describe the problem in 1 paragraph. In the second paragraph, they have to detail briefly the current solution for the problem and their lacks (the gap). The aim of the paper should be described in the third paragraph. Finally, the structure of the paper must be written in the last paragraph. A new section must be added to define the state of the art and cite the related work. This section must be added after the introduction section. Figures 1, 2 and 4 must be commented with more detail The results section must be improved. In the current result section, the authors are only testing a commercial device. This device in its datasheet is already including information about the expected errors at different temperatures and pressures. Therefore no novelty is presented in the results section. A comparison with other devices that justify why authors select this specific sensor is required. Moreover, a comparison with the current solutions (described in the related work section) will enhance the quality of the paper. In the conclusion section, future work should be included. Other minor corrections:For equation (2) a reference is needed
For controller STM32L011 a reference to its datasheet is needed. Same for MS583730BA
At the beginning of each section, a brief introduction must be included.
Reviewer 3 Report
Good and informative article showing a good implementation of existing sensors and electronic processing software and hardware for detection on level of water.
A few suggestions to improve the readability of the article:
1) In the abstract replace the second "monitoring" term by "assessment"
2) Cut the long sentence in the abstract into two sentences so that you have . "..Obtaining..."
3) page 2, replace "the practicability", "its practicability"
4) page 2, change "MS5387 by "the pressure sensor MS5837"
5) Line spacing keeps changing throughout the article, be consistent
6) Section 2.3.1. By convention, we can it sigma-delta circuit. Swap the symbols.
7) Table 2, at 10:04:00, it should be 88cm in the measured data and not 80 cm otherwise the error would not be 1 cm
8) No space between number and the units of degrees Celsius
